# Unraveling the Aquaporin-3 Inhibitory Effect of Rottlerin by Experimental and Computational Approaches

**DOI:** 10.3390/ijms24066004

**Published:** 2023-03-22

**Authors:** Inês Paccetti-Alves, Marta S. P. Batista, Catarina Pimpão, Bruno L. Victor, Graça Soveral

**Affiliations:** 1Research Institute for Medicines (iMed.ULisboa), Faculty of Pharmacy, Universidade de Lisboa, 1649-003 Lisbon, Portugal; 2Department of Pharmaceutical Sciences and Medicines, Faculty of Pharmacy, Universidade de Lisboa, 1649-003 Lisbon, Portugal; 3Biosystems and Integrative Sciences Institute, Faculty of Sciences, Universidade de Lisboa, 1649-003 Lisbon, Portugal

**Keywords:** polyphenol, aquaporin, aquaglyceroporin, glycerol permeability, inhibitor, cancer

## Abstract

The natural polyphenolic compound Rottlerin (RoT) showed anticancer properties in a variety of human cancers through the inhibition of several target molecules implicated in tumorigenesis, revealing its potential as an anticancer agent. Aquaporins (AQPs) are found overexpressed in different types of cancers and have recently emerged as promising pharmacological targets. Increasing evidence suggests that the water/glycerol channel aquaporin-3 (AQP3) plays a key role in cancer and metastasis. Here, we report the ability of RoT to inhibit human AQP3 activity with an IC_50_ in the micromolar range (22.8 ± 5.82 µM for water and 6.7 ± 2.97 µM for glycerol permeability inhibition). Moreover, we have used molecular docking and molecular dynamics simulations to understand the structural determinants of RoT that explain its ability to inhibit AQP3. Our results show that RoT blocks AQP3-glycerol permeation by establishing strong and stable interactions at the extracellular region of AQP3 pores interacting with residues essential for glycerol permeation. Altogether, our multidisciplinary approach unveiled RoT as an anticancer drug against tumors where AQP3 is highly expressed providing new information to aquaporin research that may boost future drug design.

## 1. Introduction

*Mallotus philippinensis* species are known to contain different natural compounds, mainly phenols, that were reported with a vast array of biological activities such as antimicrobial, antioxidant, antiviral, cytotoxicity, antioxidant, anti-inflammatory, immunoregulatory and anticancer activities, and have been widely used as traditional medicine in several countries [1]. Among the many natural polyphenolic compounds identified and isolated from the mature fruits of *Mallotus philippinensis*, rottlerin (RoT) has demonstrated antitumor properties in a variety of human cancers, suggesting its potential as a therapeutic agent for cancer treatments [2]. RoT was first described, and used for decades, as a potent kinase inhibitor [3,4] although several other RoT properties have also been described, such as antiproliferative [5], antiangiogenic [6], anti-inflammatory [7], anti-allergic [8], anti-microbial [9], anti-fungal [10], anti-parasitic [11] and oxidant quencher [12]. Likewise other polyphenols, RoT has antioxidant properties inhibiting ROS formation through the prevention of NF-κB activation in breast and colon cancer cells [12]. Until now, RoT has been disclosed as an agent that can regulate several targets by different mechanisms, inhibiting several signaling pathways in numerous different types of cancers [2]. While the efficacy of anticancer agents often limits their usage due to the adverse effects on normal cells, the toxicity of RoT in non-tumorigenic cells is pretty much absent, as previously demonstrated in vitro and in vivo [4,13,14,15]. Furthermore, RoT showed a number of very good drug-like pharmacokinetic properties, excellent half-life, and oral bioavailability when compared to other members of natural phenolics, making it a promising anticancer phytotherapeutic [16]. Studies of RoT anticancer activity have reported different but related mechanisms of action, including inhibition of cell growth, cell migration, and invasion, autophagy induction, triggering apoptosis, and cell cycle arrest [2], in part explaining the broad biological activity of the compound. Although described to target various proteins implicated in cellular signaling pathways, the effect of RoT in aquaporins (AQPs), membrane channels implicated in cancer progression and metastasis [17], has never been studied. AQPs are specialized transmembrane protein channels that facilitate the passive transport of water driven by an osmotic gradient, created by the active transport of solutes [18,19]. In humans, 13 aquaporins were identified so far (AQP0-12) and were found expressed in different tissues being implicated in several biological functions [19,20,21]. According to their primary sequences, structure, and permeability, AQPs were grouped into three subfamilies: (i) Orthodox or classical AQPs, which are mainly water channels (AQP0, AQP1, AQP2, AQP4, AQP5, AQP6, and AQP8); (ii) Aquaglyceroporins, which also permeate glycerol and other small uncharged solutes such as urea, in addition to water (AQP3, AQP7, AQP9, and AQP10); (iii) Unorthodox or S-aquaporins (AQP11 and AQP12) with lower primary structure homology and localized in intracellular membranes [22,23]. A fourth subfamily named peroxiporins was recently recognized, grouping AQPs also able to transport hydrogen peroxide (H_2_O_2_) [24].

AQPs were reported to be aberrantly expressed in different human tumors being implicated in cancer cell proliferation and migration, angiogenesis, and metastasis [17,25,26], strongly supporting their great potential as novel drug targets for cancer treatment [19,27]. AQP1 plays a significant role in tumor cell migration and invasion increasing cancer cell extravasation and angiogenesis [28]. Additionally, several studies show that AQP3 may also contribute to cancer progression by increasing the motility and invasiveness of cancer cells [29]. The different mechanisms proposed to explain AQP3 participation in tumor growth and spread include its ability to transport glycerol, a key molecule for metabolic reactions in energy-demanding cancer cells [17], as well as its ability to transport H_2_O_2_, modulating oxidative stress and triggering signaling cascades responsible for cell proliferation and migration [30,31]. Among the AQP-targeted drugs reported so far, metallodrugs demonstrated potent AQP3 inhibition suggesting a great potential for cancer treatment [32,33,34]. However, their associated toxicity may impair their usage in clinical trials, anticipating a broader and safer application of plant-derived bioactive compounds [35,36].

Despite all the studies characterizing RoT antioxidant and anticancer properties, its effect as a modulator of AQPs has never been explored. This prompted us to investigate the possible effect of RoT (Figure 1) on AQP1 and AQP3 activity, which are isoforms particularly implicated in tumorigenesis. Using human red blood cells (hRBCs) that highly express AQP1 [37] and AQP3 [38], we screened RoT inhibitory effect by assessing water and glycerol membrane permeability and validated its activity in yeast cells transformed with these human AQPs [39]. Moreover, to structurally understand the effect of RoT as a modulator of AQP3-mediated glycerol fluxes, we combined structural modeling and characterization with molecular docking and molecular dynamics simulations to predict structure–activity relationships of the compound’s affinity to AQP3.

## 2. Results

### 2.1. Effect of Rottlerin on AQP1 and AQP3 Activity

To assess the inhibitory effect of RoT on AQP1 and AQP3 activity, we first screened the effect of RoT on water and glycerol permeability in hRBCs, that endogenously express AQP1 (water channel [37]) and AQP3 (water/glycerol channel [38,40]) using the stopped-flow spectroscopy. To evaluate water permeability, hRBCs incubated in isotonic phosphate-buffered saline (PBS) were challenged with a hyperosmotic sucrose solution (non-permeable solute), creating an osmotic gradient, which leads to fast water efflux and subsequent cell shrinkage. For glycerol permeability, cells were challenged with a hyperosmotic glycerol solution (permeable solute), inducing a fast cell shrinkage due to water efflux followed by glycerol uptake via AQP3, resulting in water influx and cell reswelling (Figure 2A,B). Water (P_f_) and glycerol (P_gly_) permeability coefficients were calculated from the rate of cell volume change after the osmotic shock. RoT effect was obtained by treating cells with RoT (25 µM for 30 min at RT).

As shown in Figure 2C, treatment with 25 µM of RoT resulted in a 99% inhibition of glycerol permeability (P_gly_), possibly via the glycerol channel AQP3, and a 52% inhibition of water transport (P_f_) that can be mediated by both AQP1 and AQP3. To determine RoT potency, the IC_50_ values for the inhibition of water and glycerol transport were obtained through permeability assays ranging RoT concentrations from 0.1 to 150 µM. The obtained dose–response curves demonstrate a strong glycerol inhibitory potency (P_gly_ inhibition IC_50_ = 6.7 ± 2.97 µM) and a significant inhibition of water transport (P_f_ inhibition IC_50_ = 22.8 ± 5.82 µM) (Figure 2D,E).

Since the observed inhibition of hRBCs water transport may be due to the blockage of AQP1 or AQP3 or both, we used the yeast *Saccharomyces cerevisiae* model, previously optimized by our lab, and used for heterologous aquaporin functional studies [41]. Yeast cells, depleted of endogenous aquaporins, were transformed with either the empty plasmid (control cells) or the plasmid encoding the human orthodox aquaporin AQP1 (hAQP1) and the aquaglyceroporin AQP3 (hAQP3). For permeability assays, cells were loaded with the volume-sensitive dye CFDA and were challenged either with a hyperosmotic sorbitol solution or a hyperosmotic glycerol solution to evaluate water or glycerol permeability via hAQP1 and hAQP3, respectively. This enabled us to ascertain whether the decreased water permeability observed in hRBCs was due to AQP1 blockage and confirm the inhibition of glycerol and water transport mediated by AQP3. For inhibition assays, cells were previously incubated with RoT (100 µM, 30 min).

To validate the expression of hAQPs in the yeast cell transformants, their subcellular localization at the yeast plasma membrane was confirmed by fluorescence microscopy using GFP-tagging (Appendix A). Functional activity was assessed by permeability assays performed through the stopped-flow technique. As observed in Figure 3A, the measured P_f_ for hAQP1-expressing yeast cells (3.002 ± 0.428 × 10^−3^ cm s^−1^) was seven-fold higher than the empty vector (0.439 ± 0.089 × 10^−3^ cm s^−1^) and, as expected, hAQP1 did not permeate glycerol, showing P_gly_ values similar to the cells harboring the empty vector (Figure 3B). RoT did not affect water permeability in yeast-hAQP1 cells. This result indicates that the water permeability inhibition observed in hRBCs was probably due to blockage of AQP3-water fluxes. This result was confirmed by testing the water permeability of hAQP3-expressing yeast cells after treatment with RoT. As shown in Figure 3C, RoT reduced 22% the P_f_ of hAQP3 yeast cells, validating the previous data for water inhibition obtained with hRBCs.

Moreover, hAQP3-expressing yeast cells showed P_gly_ 28-fold higher than the empty vector (4.54 ± 0.331 × 10^−6^ cm s^−1^ vs. 0.162 ± 0.005 × 10^−6^ cm s^−1^) (Figure 3D). The much higher water and glycerol permeability compared with the empty vector observed in yeast transformants confirms that hAQP1 and hAQP3 are functional in yeast cell membranes, allowing to evaluate the inhibitory effect of RoT for each yeast transformant. As expected, RoT strongly impaired glycerol permeability (73% inhibition) of hAQP3-expressing yeast cells (Figure 3D), confirming RoT inhibitory activity towards AQP3.

### 2.2. Computational Studies with AQP3

Since no experimentally determined structure of AQP3 is currently available at the Protein Data Bank [42], we have initially generated a 3D model of this protein using comparative modeling to allow a thorough structural characterization of the interaction of RoT with this AQP subtype (Appendix A). To validate the selected generated model, we have produced the Ramachandran plot (Appendix A), which allows us to energetically visualize allowed regions for backbone dihedral angles ψ against φ of the amino acid residues found in the protein structure. Our results show that 92.2% of the residues of the protein are found at the most favorable regions of the Ramachandran map, except for four residues that can be found in solvent-exposed loops at the protein surface. Since these more problematic residues are found in regions with high flexibility with low functional impact, we considered that during the structure minimization and MD simulations, the problematic arrangement of these residues would fade. Our next step was to fully embed the protein in a model membrane of 1-palmitoyl-2-oleoyl-sn-glycero-3-phosphocholine (POPC) lipids. As previously described, by using the inflateGRO2 method [43] we have integrated the AQP3 homotetrameric structure on a membrane bilayer composed at the end of 382 POPC molecules (Figure 4). Since our objective was to sample the conformational space of the AQP3 homotetramer, we then performed five replicate MD simulations of the simulation system.

A high structural convergence of the AQP3 protein is observed after 200 ns of simulation in all replicate simulations (Appendix A, where we have plotted the all-atom RMSD of the full protein). These results indicate a high structural stabilization and convergence of the system in the simulated conditions. To assess the stability of the membrane properties where the protein was embedded, we have also calculated the average thickness of the bilayer throughout all replicate simulations. The overall average thickness of the POPC bilayer where AQP3 is embedded fluctuates around 3.8 nm in all replicate simulations (Appendix A), a value which is well within the experimental range determined for this type of membrane [44].

Since our goal focused on the identification of representative conformations of the AQP3 protein monomers in a binding “ready” state, we have identified two distances between four loops found at the extracellular surface of the protein that were used as references. These two distances are clearly identified in Figure 5 and are defined by: distance 1 (d1), the distance between loops composed by residues 52 to 54 and residues 209 to 211; distance 2 (d2) is the distance between loops composed by residues 144 to 151 and residues 234 to 236. By using only the equilibrated parts of all replicate simulations (200 ns onwards), we plotted the free-energy landscape of the sampled distances identified in all monomers throughout all simulation replicates, allowing the identification of the most populated conformations (defined by the basins in Figure 4). From these results, we have focused solely on the conformations that we considered to be in the “binding ready state”, which were defined by d1 and d2 values higher than 1.9 and 1.19 nm, respectively. The selection of these distances was done based on assuring that the binding region at the extracellular surface of AQP3 monomers, close to the selectivity filter, was accessible for the binding of RoT. Taking into account these considerations, we were able to identify three different basins that corresponded to conformations meeting the previously determined criteria (Figure 5A). For each identified basin (basin 1, 5 and 6), we have randomly selected five different conformations of the AQP3 monomers to perform the molecular docking calculations. In the end, we gathered 15 different AQP3 monomer conformations which were then used to dock RoT.

As shown in Figure 6, the selected AQP3 monomer configurations from basin 1 used in the docking calculations yield similar RoT docking poses. In these interaction poses, the terminal benzenotriol aromatic ring from RoT establishes consistent hydrogen bond interactions with serine 152, found in one of the loops at the extracellular face of AQP3 protein, close to the access of the pore. Furthermore, the central chromenol derivative group with the two hydroxyl and aromatic properties seems to occupy a more buried position at AQP extracellular surface, by stacking with aromatic residues such as valine 43 and valine 47. Additionally, the phenylpropyl chemical group, due to its conformational flexibility, seems to provide additional stabilization of the interaction of the compound with the multiple hydrophobic and aromatic residues found in this region of the protein. The reported interaction profile is consistent between the different used AQP3 monomer configurations from basins 5 and 6 (Appendix A).

To evaluate the dynamics of interaction established between RoT and the AQP3 binding site, and the convergency of the docking poses previously reported, we have performed MD simulations of the 15 different complexes previously determined. These interaction configurations were re-integrated in the configurations of the AQP3 homotetrameric system previously simulated and used as the starting points for the MD simulations of RoT and AQP3 complexes. As shown in Appendix A, where we have represented the RMSD of RoT throughout all the simulations with respect to the initial position determined on basin 1 replicate 1 conformation, we can see that a wide range of RMSD values is obtained (ranging from 0.1 nm to 0.35 nm). Interestingly, in all the simulations, RoT remains bound to AQP3 in similar positions when compared to the initially determined docking positions, evidencing the formation of highly stable complexes. Despite the high affinity of RoT to the AQP3 binding site, a specific binding pose was somehow difficult to identify. According to our results, the high flexibility of RoT allows it to easily adapt to the highly flexible extracellular region at the surface of AQP3, close to the selectivity filter.

Figure 7 shows the residues on AQP3 that most contributed to the interaction with RoT throughout all equilibrated parts of the MD simulations. Tyr 150 and Phe 208 are the two residues that establish consistent interactions with RoT. These two residues have, respectively, hydrophilic and hydrophobic characteristics that establish a stable interaction with both the benzenotriol and chromenol groups of RoT.

As can be seen in Figure 8, where we have represented a highly populated and representative conformation of RoT, while the benzenotriol group is highly exposed to the solvent molecules, the chromenol group is found deeper at the regions close to the entrance of the pore. Interestingly, and in agreement with what was previously observed in the molecular docking results, the phenylpropyl group seems to further boost the interaction of RoT by directly interacting with the more hydrophobic residues found at the surface of the protein. By establishing highly stable complexes with the extracellular region of AQP3, RoT acts as a stereochemical lid that highly inhibits the reported glycerol permeability through the pores of the protein. However, the established interaction configurations are labile enough to allow the permeation of water molecules, which are individually smaller and more dynamic when compared to glycerol. These results agree with the experimental information withdrawn from the above-reported data, where the IC_50_ for the inhibition of water permeability is higher than for glycerol.

## 3. Discussion

RoT is currently considered a multitarget drug that can be cytotoxic for different cancer cell types having a successful anticancer action [4,45,46]. Different studies suggested that RoT can inhibit tumor progression at several levels and in different cell models, inducing apoptosis, triggering cell cycle arrest, retarding cell migration and invasion in various types of human cancer [2] being an efficient compound against different tumor cells with multiple mechanisms of action [4], including inhibition of protein kinases with some specificity for Protein Kinase C [3]. The wide spectrum of activities makes RoT a promising compound for novel anticancer therapeutic approaches [4].

Aquaporins have been associated with cancer invasion, and metastasis [25] and, in particular, AQP3 seems to affect cellular functions commonly associated with cancer progression, including proliferation, migration and metastasis [19,29,30,47,48]. The current study reveals for the first time the ability of RoT to inhibit AQP3 activity, an aquaporin with a key role in tumor growth and spread by mechanisms still under investigation. Therefore, targeting AQP3 is trusted to have great potential for cancer treatments [49]. In this sense, phytocompounds emerged as modulators of AQPs expression and function with anticipated clinical benefits [35,36].

This study reveals the inhibitory effect of RoT on AQP3, impairing water and glycerol membrane permeability of hRBCs, with an IC_50_ in the micromolar range. The higher IC_50_ value for water permeability compared to glycerol could account for the lower reduction in water permeability induced by RoT. These results were further validated in yeast cells transformed with human AQP1 and AQP3 and revealed the selectivity of the compound for AQP3 with no measurable effect on AQP1. The confirmed AQP3 expression and localization at the cell plasma membrane previous to functional assays, indicates that RoT has a direct effect on AQP3 channel activity with no interference in upstream protein translocation mediated by Protein Kinase C.

The possible mechanism of RoT inhibition was investigated by computational approaches, using a generated 3D model of the protein that was further validated, followed by molecular docking studies to predict the most plausible binding configurations. According to our computational predictions, RoT establishes strong and consistent interactions with several residues found at the extracellular surface region of AQP3, acting as a stereochemical cap that blocks glycerol from accessing the pores of the protein. Both the benzenotriol and chromenol chemical groups confer the required stability of RoT to act as a lid, which is further boosted by the high flexible phenylpropyl that established strong interactions with several hydrophobic residues around the aromatic filter at the pore entrance facing the extracellular medium. The strong binding of RoT at the extracellular surface, close to the selectivity filter, explains its high affinity that prevents glycerol from permeating through the AQP3 protein, although some permeation of water molecules may still occur.

In conclusion, we were able to demonstrate that RoT affects AQP3 function, and this effect can also explain the vast anticancer activity reported for this polyphenol compound. Our calculations allowed a thorough structural characterization of the interaction between RoT and AQP3, evidencing vital structural groups essential for binding. Further studies to validate the herein proposed binding mechanism are paramount for the future development of RoT derivatives with improved affinity to AQP3, potentiating its anticancer therapeutical application. Overall, our study adds new knowledge to the aquaporin research field and highlights RoT as a promising natural lead compound for the development of anticancer therapeutics for tumors where AQP3 is highly expressed.

## 4. Materials and Methods

### 4.1. Rottlerin

The compound Rottlerin (RoT) was obtained from a commercial database (Sigma Aldrich, St. Louis, MO, USA) and was solubilized in DMSO to 10 mM final concentration (stock solution).

### 4.2. Ethics Statement

Venous blood samples were obtained from healthy human volunteers following a protocol approved by the Ethics Committee of the Faculty of Pharmacy of the University of Lisbon (Instituto Português de Sangue Protocol SN-22/05/2007). Informed written consent was obtained from all participants.

### 4.3. Red Blood Cells Sampling and Preparation

Venous blood samples were collected from anonymous human donors in citrate anticoagulant (2.7% citric acid, 4.5% trisodium citrate, and 2% glucose). Red blood cells were isolated by centrifugation (750× *g*, 10 min) at room temperature (RT) and washed with PBS (NaCl 137 mM, KCl 2.7 mM, Na_2_HPO_4_ 10 mM, KH_2_PO_4_ 1.8 mM, pH 7.4). Cells diluted in PBS (0.5% suspension) were immediately used in the assays.

### 4.4. Yeast Strain and Growth Conditions

Saccharomyces cerevisiae 10560-6B MATα leu2::hisG trp1::hisG his3::hisG ura352 aqy1D::KanMX aqy2D::KanMX (YSH1770) was used as a host strain for heterologous expression of human AQP1 and AQP3. To transform the yeast cells, the complementary DNA (cDNA) from human aquaporins (AQP1 and AQP3) was amplified by PCR from the pWPi-DEST plasmid [50] and C-terminally fused to a green fluorescent protein (GFP) of the centromeric plasmid, pUG35 [51]. *Escherichia coli* DH5α was used as a host for routine propagation and plasmids were purified with a GenEluteTM Plasmid Miniprep Kit (Sigma-Aldrich, St. Louis, MO, USA). *Escherichia coli* transformants were maintained and grown in Luria–Bertani broth (LB) with ampicillin (100 µg mL^−1^) at 37 °C [52].

Yeast cultures were grown at 27 °C with orbital shaking in yeast nitrogen base (YNB) without amino acids (DIFCO), with 2% (*w*/*v*) glucose and supplemented with adequate requirements for prototrophic growth [53]. Transformants were grown to OD_600 nm_ ≈ 1, harvested by centrifugation (5000× *g*, 10 min, 4 °C), washed three times and re-suspended in ice-cold sorbitol (sorbitol 1.4 M in 50 mM K^+^ citrate buffer, pH 7.4) up to a concentration of 0.33 g (wet weight) mL^−1^ and kept on ice for at least 90 min. Prior to osmotic challenges, cells were preloaded with the nonfluorescent precursor 5-(and6)-carboxyfluorescein diacetate (CFDA, Sigma, St. Louis, MO, USA; 1 mM for 20 min at 30 °C) that is cleaved intracellularly by nonspecific esterases to generate the membrane-impermeable fluorescent form carboxyfluorescein (CF) known to remain in the cytoplasm [38]. Cells were then diluted 1:10 in 1.4 M sorbitol buffer and immediately used for stopped-flow experiments.

The localization of GFP-tagged AQP1 and AQP3 in the yeast cells could be confirmed with a Zeiss Axiovert 200 fluorescence microscope, at 495 nm excitation and 535 nm emission wavelengths, assuring more than 80% localization at the yeast plasma membrane.

### 4.5. Permeability Assays

Light scattering and fluorescence stopped-flow spectroscopy were used to monitor cell volume changes of hRBCs [54] and yeast transformants loaded with the concentration-dependent self-quenching fluorophore 5-(and-6)-carboxyfluorescein diacetate (CFDA, 1 mM, 20 min at 30 °C) [41], respectively.

Experiments were performed on a HI-TECH Scientific PQ/SF-53 stopped-flow apparatus, with a 2-ms dead time and temperature-controlled, interfaced with an IBM PC/AT compatible 80,386 microcomputer. After challenging cell suspensions with an equal volume of shock solution at 23 °C, the time course of volume change was measured by following the 90° scattered light intensity at 400 nm or fluorescence intensity (excitation 470 nm and emission 530 nm). For each experimental condition, 5–7 replicates were analyzed. Baselines were acquired using the respective incubation buffers as isotonic shock solutions. For osmotic water permeability (P_f_) measurements, a hyperosmotic shock solution containing a non-permeable solute was used (for RBC assays, sucrose 200 mM in PBS pH 7.4; for yeast assays, sorbitol 2.1 M in K^+^ citrate pH 7.4) producing an osmotic gradient that results in water efflux and consequent cell shrinkage. For glycerol permeability (P_gly_) measurements, a hyperosmotic shock solution containing a permeable solute was used (for RBC assays, glycerol 200 mM in PBS pH 7.4; for yeast assays, glycerol 2.1 M in K^+^ citrate pH 7.4) creating an inwardly directed glycerol gradient. After the first fast cell shrinkage due to water efflux, glycerol influx in response to its chemical gradient is followed by water with subsequent cell re-swelling.

For hRBCs, P_f_ was estimated by P_f_ = k (V_o_/A)(1/V_w_ (osm_out_)_∞_) (cm s^−1^), where V_w_ is the molar volume of water, V_o_/A is the initial cell volume to area ratio, (osm_out_)_∞_ is the final medium osmolarity after the applied osmotic gradient and k is the single exponential time constant fitted to the light scattering or fluorescence signal of hRBCs shrinkage [54]. P_gly_ was calculated by P_gly_ = k (V_o_/A) (cm s^−1^), where V_o_/A is the initial cell volume-to-area ratio, and k is the single exponential time constant fitted to the light scattering signal of glycerol influx in erythrocytes [40].

For yeast cells, the fluorescent glycerol traces obtained were corrected by subtracting the baseline slope that reflects the bleaching of the fluorophore. Optimization of yeast permeability values was accomplished by numerical integrations using a mathematical model implemented in the Berkeley Madonna software (version 10.2.8) as described [52].

To assess the effect of RoT, cells were incubated with different concentrations of the compound for 30 min at RT before stopped-flow experiments. The inhibitor concentration that induces 50% inhibition (IC_50_) was calculated by nonlinear regression of dose–response curves (GraphPad Prism software) using the following equation: y = 100/(1 + 10^((LogIC50-X)×HillSlope)^); where HillSlope describes the steepness of the family of curves.

### 4.6. Statistical Analysis

All the experiments were performed in three biological and at least three technical triplicates. Data are presented as mean ± standard deviation (SD). Statistical analysis between groups was performed by two-way ANOVA and non-parametric Mann–Whitney test. A *p*-value < 0.05 was considered statistically significant. Statistical analysis was performed using Graph Prism software (version 9.0).

### 4.7. Modeling the Tetrameric Structure of AQP3

Since no experimental structure for AQP3 homotetramer is currently resolved, we have used a comparative modeling approach to generate a 3D model of this target protein. To identify a template structure that we could use as a reference to generate a 3D model of AQP3, we ran a BLAST [55] on UniProt with the blosum62 matrix. The PDB structure showing the highest amino acid sequence identity with AQP3 (50.2%) was the one with PDB id equal to 6F7H from AQP10 [56]. We then used the MODELLER package (version 10.1) [57] to initially generate a sequence amino acid alignment of AQP3 with AQP10, and afterward generate 1000 different models of AQP3 protein monomer. The best model was chosen based on the best DOPE score (−35,487.91406), whose structure was validated with the procheck software to produce the respective Ramachandran plot [58] (see Appendix A). The first 16 residues on the N-terminal and the last 25 residues on the C-terminal of the protein monomer corresponded to clear disorder motifs. Therefore, we truncated these residues with unidentified functions to minimize possible high fluctuation in our simulations. The full homotetramer AQP3 structure was then reconstructed based on AQP10 homotetrameric structure, by aligning the generated AQP3 monomer, to each one of the four monomers from the template structure. In the end, the full tetrameric structure of AQP3 was obtained (Appendix A).

### 4.8. Assemble the Simulation System and Sample the Conformational Space of AQP3

To integrate the chosen model on a pure POPC bilayer membrane, we used the InflateGRO2 method [43,59]. This approach employs lateral lipid translation within the membrane plane using scaling factors to first expand the membrane and then, delete lipids within a user-defined distance cutoff around the protein, which in this work was set to 5 Å. In the end, a series of alternating steps of compression and energy minimization brings the system back to its natural dimensions. In the end, the AQP3 tetrameric structure was assembled in a membrane bilayer previously equilibrated with 382 POPC lipids [60]. Once the protein was fully embedded in the bilayer membrane, the system was solvated with 36,237 SPC water molecules [61]. All titrating residues of the protein were assigned with an initial protonation state determined at physiological pH, based on their pKa, determined using the pyPKA server [62]. No ions were required to add to the simulation’s system since it was electroneutral. Five replicate Molecular dynamics (MD) simulations were carried out to sample the conformational space of the AQP3 and allowed the identification of representative conformations to use on the molecular docking calculations with RoT. The simulations were carried out with GROMACS 2018.6 package [63] and the GROMOS 54A7 force field [64,65]. The system conformational space was sampled in an NPT ensemble where the pressure (1 bar with a coupling constant of 2 ps) and temperature (298.15 K with a coupling constant of 0.1 ps) were kept constant using the Parrinello–Rahman barostat [66,67] and the v-rescale thermostat [68], respectively. The electrostatic interactions were accounted for with the Particle mesh Ewald (PME) method with a real space cut-off of 1.0 nm and a Fourier grid spacing of 0.12 nm [69], while the van der Waals interactions were truncated above 1.0 nm. All bonds were constrained using the P-LINCS algorithm [70] for membrane and solutes, and SETTLE for water [71]. The equations of motion were integrated every 2 fs with the neighbor lists being updated every 10 steps. A minimization and initiation protocol was performed in all systems to avoid unfavorable interactions. In the minimization of the system, we used the steepest descent algorithm [72] in two distinct steps. In the first position, restraints were applied to all heavy atoms of the AQP3 protein, while in the second, no restraints were applied. The initiation step was also performed with a two-step strategy of 100 ps each: the first was performed at constant volume and the second with a temperature coupling constant of 0.01 ps to avoid large fluctuations. All replicate simulations were then performed for 500 ns. The initial 200 initial periods were discarded from the analysis to allow a preliminary adequate equilibration of the simulation system.

MD simulations of the AQP3 protein complexed with RoT were also performed to evaluate the dynamics of the interaction of the compound with AQP3. The topology of RoT was obtained from the Automated Topology Builder and Repository [73,74], and manually curated to include the complete 1–4 exclusions in rings as previously done in other works [60,75]. The starting conformations for this set of simulations were obtained from the Molecular Docking calculations as explained in the next section. In the end, 15 different initial docking configurations were sampled for 200 ns. For this set of simulations, the same previously described workflow and simulation conditions used to simulate AQP3 fully embedded in the POPC membrane bilayer were followed.

### 4.9. Molecular Docking

All molecular docking calculations were performed using AutoDockFR [76]. This program has a newly implemented docking engine based on the AutoDock4 scoring function, coupling a new genetic algorithm and a customized scoring function. Initially, the required RoT and selected AQP3 conformations pdbqt files were generated using, respectively, the prepare_ligand4.py and the prepare_receptor4.py scripts found on the AutoDockTools software package (version 1.5.7) [77]. While RoT was set to be fully flexible, the selected AQP3 conformations were set to their fixed configurations. The affinity maps for the atom types found on RoT were calculated using the available agfr script on the AutoDockFR software package. The default space grid mesh for all grid maps was set to 0.375 Angstroms and was centered on the C-alpha atom of alanine 203, found at the extracellular region of the monomers of interest. The size of the search space was set to 31, 32, and 23 dimensions on the X, Y, and Z axis to assure that all the surface of AQP3 protein facing the extracellular interface was explored. Each docking run was performed using the adfr script available in the AutoDockFR software package, by setting the genetic algorithm evolutions to 2000 and the number of maximum evaluations to 25,000,000. All the remaining parameters in the search mode were set to their auto option. In the end, a 2 Angstroms range was used to generate the final clusters. For each docking run, the lowest energy solution of the top-ranking cluster was selected as the starting configuration for the MD simulations of RoT in a complex with AQP3. In the end, 15 different docking poses were used as starting configurations for the MD simulations of RoT complexed with AQP3 protein.

## Figures and Tables

**Figure 1 ijms-24-06004-f001:**
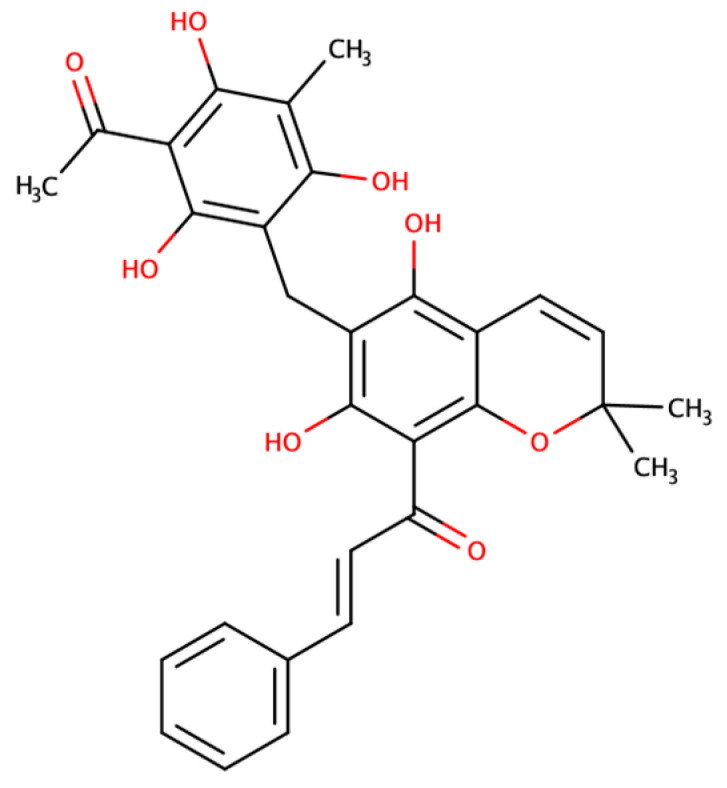
Chemical structure of Rottlerin (RoT).

**Figure 2 ijms-24-06004-f002:**
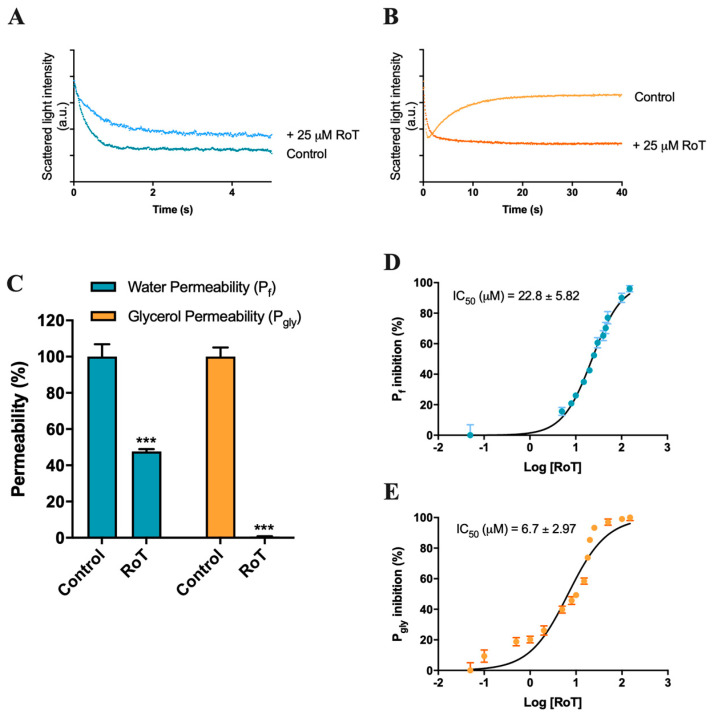
Effect of RoT on hRBCs membrane permeability. (**A**) Representative stopped-flow signal of cells confronted with a hyperosmotic sucrose solution. Cells shrink due to water efflux (control). Cell treatment with RoT decreases water efflux rate. (**B**) Representative stopped-flow signal of cells are confronted with a hyperosmotic glycerol solution. After a first cell shrinkage due to water efflux, glycerol enters the cell via AQP3, resulting in water influx and cell reswelling (control). Cell treatment with RoT prevents glycerol influx. (**C**) Water and glycerol permeability of cells incubated with RoT (25 µM for 30 min). (**D**) Dose-response curve of water permeability inhibition by RoT. (**E**) Dose-response curve of glycerol permeability inhibition by RoT. Data are shown as mean ± SD of three independent experiments. *** *p* < 0.001, treated vs. non-treated cells.

**Figure 3 ijms-24-06004-f003:**
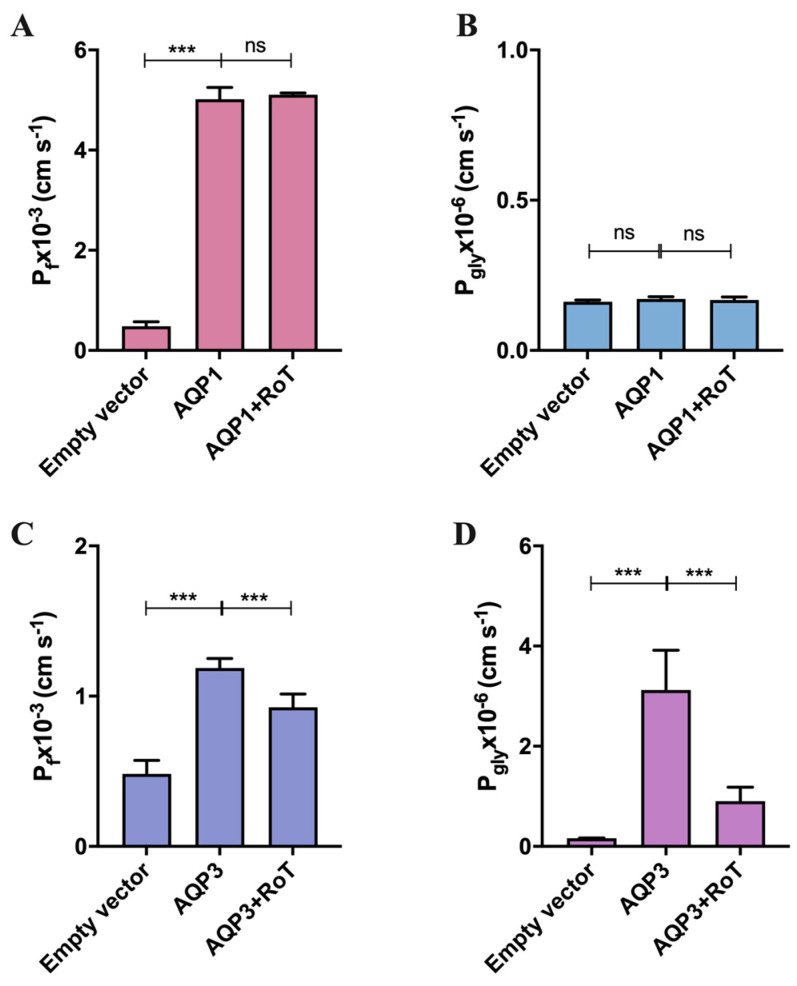
(**A**) Water permeability (P_f_) and (**B**) glycerol permeability (P_gly_) of yeast cells transformed with the empty vector, and cells expressing hAQP1 before and after treatment with RoT (100 µM); (**C**) Water permeability (P_f_) and (**D**) glycerol permeability (P_gly_) of yeast cells transformed with the empty vector, and cells expressing hAQP3 before and after treatment with RoT (100) µM. Data are shown as mean ± SD of three independent experiments. *** *p* < 0.001, ns, non-significant; empty vector vs. non-treated vs. treated cells.

**Figure 4 ijms-24-06004-f004:**
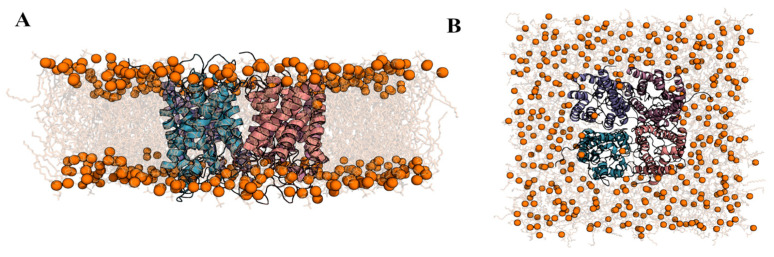
Structural representation of the initial conformation of the homotetramer of AQP3 fully embedded in a model POPC membrane. (**A**,**B**) represent, respectively, the side and top view of the simulation system. Each monomer of the AQP3 protein is colored differently as cartoon, while the phosphorous atoms of the POPC lipids are represented as orange spheres, attached to the lipidic tails, and which are represented with grey sticks. Water molecules are not explicitly represented in the figure.

**Figure 5 ijms-24-06004-f005:**
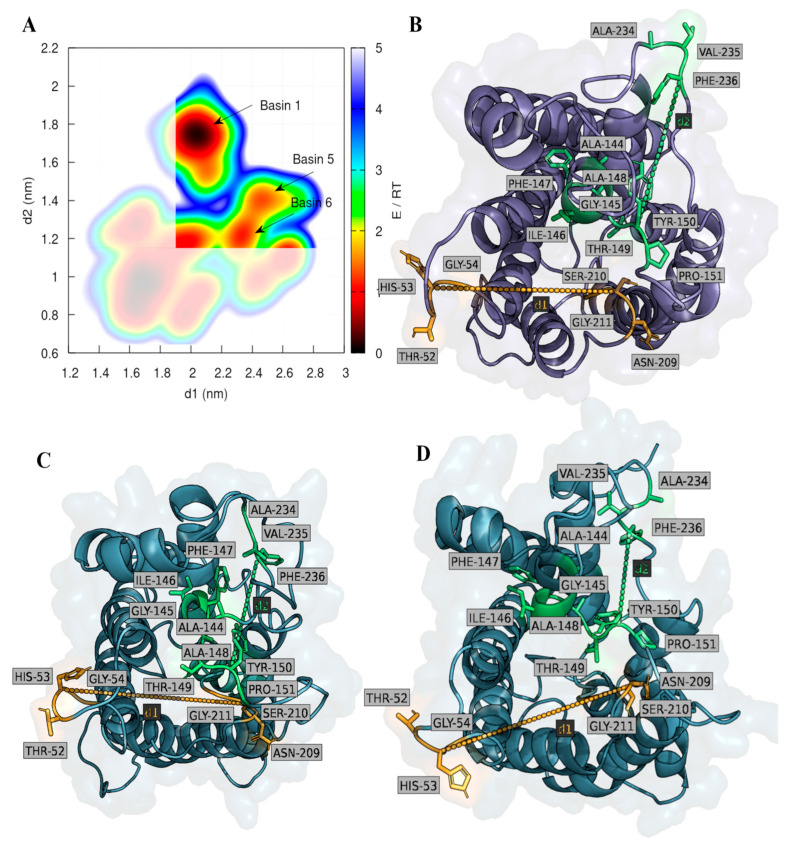
(**A**) Free-energy landscape of d1 and d2, calculated from the conformational sampling of the AQP3 system fully embedded in a POPC membrane bilayer. (**B**–**D**) highlight structural representative configurations of AQP3 monomers extracted from basins 1, 5, and 6, respectively. In these figures, d1 and d2 used as references are highlighted in yellow and green, respectively.

**Figure 6 ijms-24-06004-f006:**
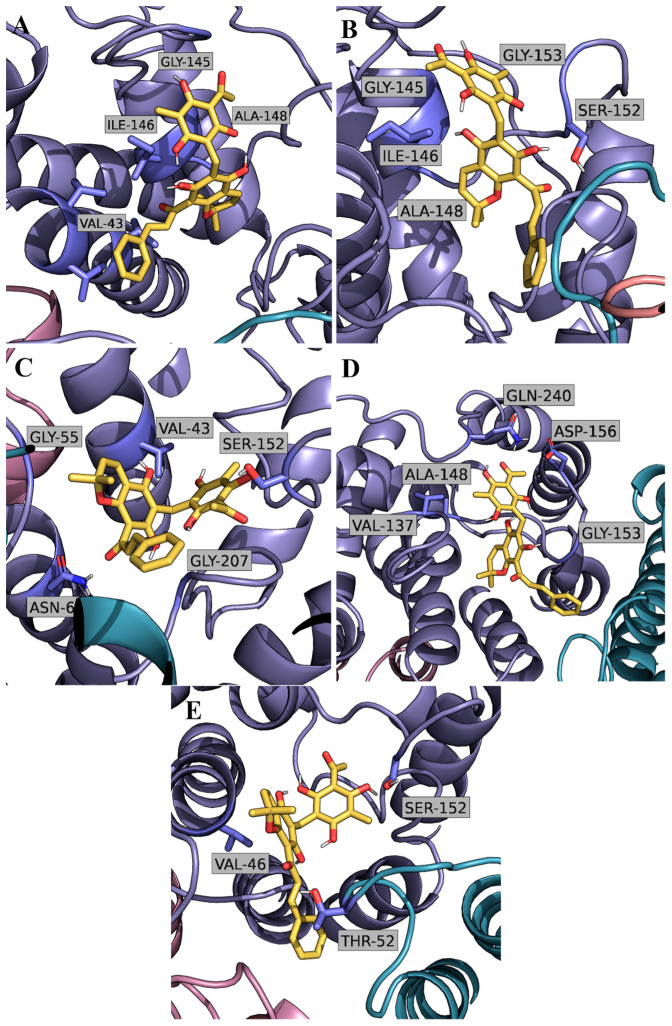
Representative molecular docking poses of RoT on the binding region of AQP3 taken from the used representative configurations from basin 1. (**A**–**E**) subfigures highlight the interactions established between RoT and AQP3 residues.

**Figure 7 ijms-24-06004-f007:**
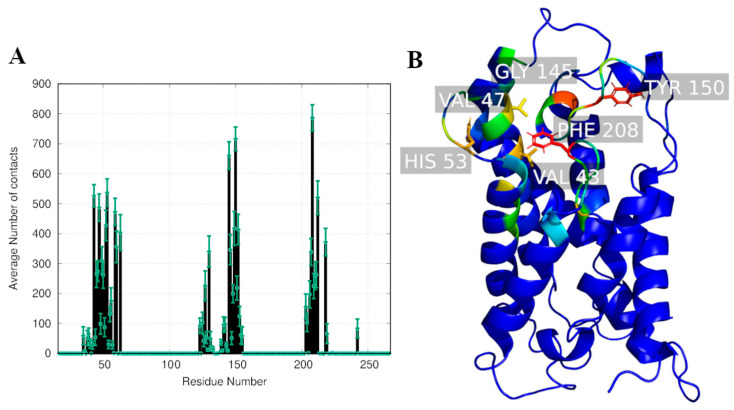
(**A**) Average number of contacts established between RoT and AQP3 residues throughout all the MD replicate simulations (between 100 ns to 200 ns of simulation). (**B**) Identification of the residues with higher number of contacts established between RoT and AQP3. The color code used ranges between red (highest number of contacts) to dark blue (lowest number of contacts).

**Figure 8 ijms-24-06004-f008:**
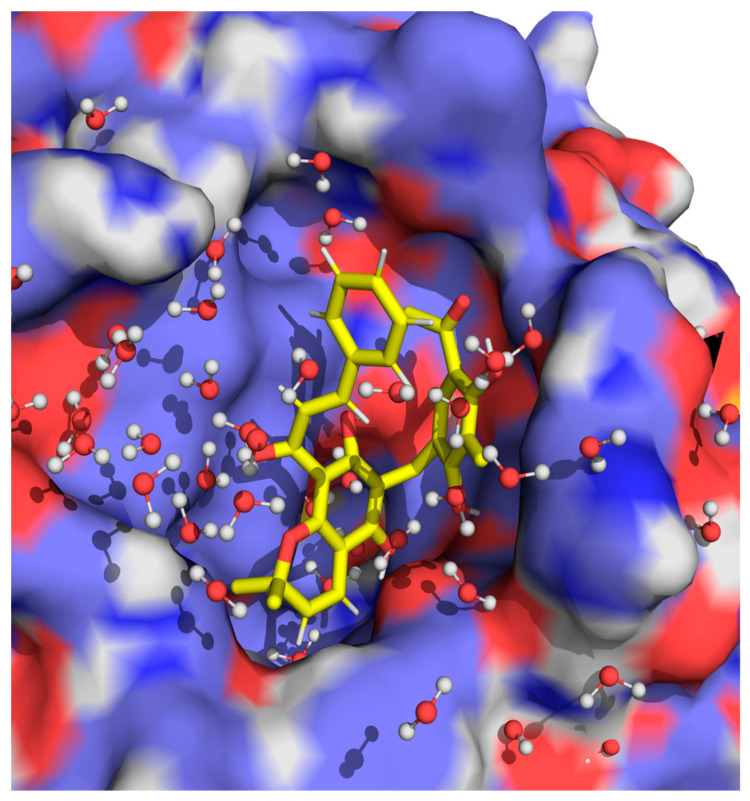
Representation of the spatial region typically occupied by RoT when interacting with the extracellular surface of AQP3, near the selectivity filter.

## Data Availability

Not applicable.

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
