# Peer review of "Unraveling the Aquaporin-3 Inhibitory Effect of Rottlerin by Experimental and Computational Approaches"

_ijms, 2023, doi:10.3390/ijms24066004_

Round 1
Reviewer 1 Report
In this manuscript, Paccetti-Alves et al. found the natural polyphenolic compound Rottlerin (RoT) could inhibit the activity of human AQP3; moreover, the authos used molecular docking and molecular dynamics simulations to reveal that RoT blocks AQP3-glycerol permeation by interacting with amino acid residues Tyr 150 and Phe 208 of AQP3. To support this conclusion, the two residues Tyr 150 and Phe 208 should be mutated on AQP3 and the inhibtory effect of RoT be observed to verify the MD simulation predicted interactions between RoT and AQP3.
Lines 108-115: To reach the conclusions made in this paragraph, the authors should provide evidences to demonstrate that AQP1 and AQP3 are the only expressed AQPs in hRBCs.
Lines 152-157: The pf of AQP3 for water and the inhibitory effect of RoT on water permeation of AQP3 should be measured to support the fact "water permeability inhibition observed in hRBCs might be due to blockage of AQP3-water fluxes". In addition, AQP1 permeability for glycerol might also be measured as a control.
Lines 126-130 and 351-352: The fluorescent pictures reflecting the localization of GFP-tagged AQP1 and AQP3 in the yeast cells should be given.
Lines 248-250: Two key amino acid residues Tyr 150 and Phe 208 should be mutated on AQP3 to verify the MD simulation predicted interactions of RoT and AQP3.
Author Response
Thank you for the careful evaluation of our manuscript. Please see below the replies to the raised questions.
1. In this manuscript, Paccetti-Alves et al. found the natural polyphenolic compound Rottlerin (RoT) could inhibit the activity of human AQP3; moreover, the authos used molecular docking and molecular dynamics simulations to reveal that RoT blocks AQP3-glycerol permeation by interacting with amino acid residues Tyr 150 and Phe 208 of AQP3. To support this conclusion, the two residues Tyr 150 and Phe 208 should be mutated on AQP3 and the inhibtory effect of RoT be observed to verify the MD simulation predicted interactions between RoT and AQP3.
Reply: Thank you for the careful evaluation of our manuscript. This work is the first evaluation and initial characterization of the activity of RoT as an inhibitor of Aquaporins, which was evidenced by functional assays and by computational studies, suggesting that the residues Tyr 150 and Phe 208 might be important for the compound stabilisation and protein channel blockage. However, for this paper we do not have time and means to perform all the needed mutagenesis studies within the requested time frame. This will be certainly validated in our future study, aiming to investigate the mechanisms of RoT inhibition to ascertain structure activity relationships.
2. Lines 108-115: To reach the conclusions made in this paragraph, the authors should provide evidences to demonstrate that AQP1 and AQP3 are the only expressed AQPs in hRBCs.
Reply: The transport of water and glycerol across the erythrocyte membrane, for its direct physiological significance and for its role in model studies, has been extensively investigated. Two AQPs are natively expressed on human erythrocytes (hRBCs): the water-conducting aquaporin AQP1 (Preston et al, Science 256, 385–387, 1992, doi: 10.1126/science.256.5055.385) and the water-and-glycerol-conducting aquaglyceroporin AQP3 (Roudier et al., J Biol Chem 273, 8407–8412, 1998, doi: 10.1074/jbc.273.14.8407). AQP1 is abundantly expressed in the human erythrocyte for maintaining its hydrohomeostasis where AQP3 is also expressed (at a level ~30-folds lower than AQP1) facilitating glycerol transport. Until now, no other AQPs were reported in human RBCs. Several studies in the literature have used hRBCs to detect the activity of AQP1 (water transport) and/or AQP3 (glycerol transport). Among many others, these are examples of such studies: Martins et al, ChemMedChem 8, 1086-1092, 2013; doi: 10.1002/cmdc.201300107; Rodriguez et al., Biochim Biophys Acta Biomembr. 1861(4): 768–775, 2019, doi: 10.1016/j.bbamem.2019.01.008; Benga, European Biophysics Journal, 42(1): p. 33–46, 2013, doi: 10.1007/s00249-012-0868-7; Martins et al., PLoS ONE, 2012. 7(5): p. E37435. doi: 10.1371/journal.pone.0037435; Campos et al., Biochemical and Biophysical Research Communications, 408(3): p. 477–481, 2011, doi: 10.1016/j.bbrc.2011.04.057.
3. Lines 152-157: The pf of AQP3 for water and the inhibitory effect of RoT on water permeation of AQP3 should be measured to support the fact "water permeability inhibition observed in hRBCs might be due to blockage of AQP3-water fluxes". In addition, AQP1 permeability for glycerol might also be measured as a control.
Reply: Thank you for the suggestion. We performed new experiments and expanded the information in Figure 3 to include the effect of RoT on water permeation of AQP3 and on the permeability to glycerol of AQP1. As expected, AQP1 did not permeate glycerol (it is reported as a strict water channel and classified as orthodox aquaporin, Preston et al, Science 256, 385–387, 1992, doi: 10.1126/science.256.5055.385). AQP3 is a water-glycerol channel, whose water permeability was reduced by 22% by RoT. These results confirmed our previously observed inhibition of water permeability in hRBCs. These data are included in the new Figure 3B and C.
4. Lines 126-130 and 351-352: The fluorescent pictures reflecting the localization of GFP-tagged AQP1 and AQP3 in the yeast cells should be given.
Reply: According to the reviewer's suggestion, we have included a new Figure S1 in Supplemental Material, showing the GFP-tagged AQP1 and AQP3 in transformed yeast cells.
5. Lines 248-250: Two key amino acid residues Tyr 150 and Phe 208 should be mutated on AQP3 to verify the MD simulation predicted interactions of RoT and AQP3.
Reply: We thank the comment of the reviewer, and we agree that performing the mutation of these two residues could confirm the specificity of the binding of RoT to AQP3. To get relevant conclusions these simulations should be performed independently and simultaneously using alanine residues and other residues with different physicochemical properties. However, to perform all these simulations with the required simulation sampling (as used in this work), we would need at least 6 months of additional simulations, making it impossible to be accounted for in this work. We are currently planning to perform all these simulations, together with experimental mutagenesis, in a future study investigating mechanisms of selectivity and structure activity relationships of RoT and derivatives.
Reviewer 2 Report
The paper interestingly describes the aquaporin-3 inhibitory effect of rottlerin by experimental and computational approaches. However, the study would benefit of performing possible additional experiments to further support the conclusion, and of discussing more in depth about the limitations of their findings.
Using human red blood cells, its was shown that 25µM Rottlerin inhibited by 99% the glycerol permeability and by 52% the water permeability. Furthermore, the IC50 of Rottlerin on glycerol and water permeabilities were 6.7 ± 2.97 µM and 22.8 ± 5.82 µM. In the discussion section, the authors should indicate that the lower IC50 of Rottlerin on water permeability could account for the lower reduction in water permeability observed with 25µM Rottlerin. It could be interested to determine the effects of Rottlerin derivatives on hAQP1 and hAQP3 permeabilities to determine their potency and efficacy, as well as to test them using molecular docking and molecular dynamics simulations.
To determine if the inhibition of glycerol and water permeabilities in human red blood cells is due to the blockage of AQP1 or AQP3 or both, the water permeability of yeast expressing hAQP1 and the glycerol permeability of yeast expressing hAQP3 was measured in the absence or presence of 100µM Rottlerin. Considering the IC50 values of Rottlerin on water and glycerol permeabilities, a smaller concentration of Rottlerin could be used to inhibit the glycerol permeability. Furthermore, considering lines 152-154 “As observed in Figure 3A, Rot did not affect permeability in yeast-hAQP1 cells. This results indicated that the water permeability inhibition observed in hRBCs might be due to blockage of AQP3-water fluxes”: was the water permeability of AQP3 inhibited by Rottlerin and if so to which extent? Were the IC50 of Rottlerin on water permeability of hAQP1 and hAQP3 and on glycerol permeability of hAQP3 determined using dose-response curves? Are the doses of Rottlerin used to inhibit AQPs permeabilities in red blood cells and yeast compatible with cell usage (are they inducing any cell mortality) and do they inhibit protein kinase C activity as well? Does Rottlerin affect the water and glycerol permeabilities of other AQPs?
Protein kinase C has been involved in cell volume regulation (e.g. PMIDs: 9755245, 8038218, 1560240) as well as phosphorylation and translocation of AQP1 (PMID: 22334691). Beyond the computational analysis, what are the experimental evidence for a direct effect of Rottlerin on hAQP3, rather than an indirect effect via its inhibitory action on protein kinase C that may be stimulated upon osmotic challenge? What are the effects of other protein kinase C inhibitors on the hAQP1 and hAQP3 water and glycerol permeabilities? Therefore, in cancers, the authors should discuss if the effects of Rottlerin may be attributable to protein kinase C and/or AQP3 inhibition, taking into account protein kinase C has also been involved in cancers.
Author Response
Thank you for the careful evaluation of our manuscript. Please see below the replies to the raised questions.
1. The paper interestingly describes the aquaporin-3 inhibitory effect of rottlerin by experimental and computational approaches. However, the study would benefit of performing possible additional experiments to further support the conclusion, and of discussing more in depth about the limitations of their findings. Using human red blood cells, its was shown that 25µM Rottlerin inhibited by 99% the glycerol permeability and by 52% the water permeability. Furthermore, the IC50 of Rottlerin on glycerol and water permeabilities were 6.7 ± 2.97 µM and 22.8 ± 5.82 µM. In the discussion section, the authors should indicate that the lower IC50 of Rottlerin on water permeability could account for the lower reduction in water permeability observed with 25µM Rottlerin.
Reply: As suggested, in the discussion we added this information in the discussion: “The higher IC50 value for water permeability compared to glycerol could account for the lower reduction of water permeability induced by RoT”.
2. It could be interested to determine the effects of Rottlerin derivatives on hAQP1 and hAQP3 permeabilities to determine their potency and efficacy, as well as to test them using molecular docking and molecular dynamics simulations.
Reply: Thank you for the relevant comment. This work is the first evaluation and initial characterization of the activity of RoT as an inhibitor of Aquaporins. The experimental derivatization, simulation, and structure-activity characterization of the interaction between RoT derivatives and AQPs is our next goal, and will be the explored in a future work. We have added a sentence in the conclusions addressing the need of future investigation of RoT derivatives with improved affinity to AQP3, in order to potentiate its anticancer therapeutical application.
3. To determine if the inhibition of glycerol and water permeabilities in human red blood cells is due to the blockage of AQP1 or AQP3 or both, the water permeability of yeast expressing hAQP1 and the glycerol permeability of yeast expressing hAQP3 was measured in the absence or presence of 100µM Rottlerin. Considering the IC50 values of Rottlerin on water and glycerol permeabilities, a smaller concentration of Rottlerin could be used to inhibit the glycerol permeability.
Reply: We agree that a smaller concentration could have been used in yeast assays. However, since the IC50 values were obtained with hRBCs and from our experience the values are sometimes higher in yeast cells due to differences in membranes and in protein levels (Mósca et al. Cells 7, 207, 2018, doi:10.3390/cells7110207), we decided to use a larger concentration of RoT to ensure that any inhibition of AQP1 would be detected. Using this higher concentration we were able to demonstrate the lack of RoT effect on AQP1 and to validate its selectivity towards AQP3.
4. Furthermore, considering lines 152-154 “As observed in Figure 3A, Rot did not affect permeability in yeast-hAQP1 cells. This results indicated that the water permeability inhibition observed in hRBCs might be due to blockage of AQP3-water fluxes”: was the water permeability of AQP3 inhibited by Rottlerin and if so to which extent?
Reply: We performed new experiments that are shown in Fig.3C, showing the water permeability of hAQP3-yeast cells. Indeed, a inhibition of 22% of Pf (water permeability) of AQP3 was detected, corroborating the previously described data for hRBCs. This new data and respective figure (Fig.3C) were added to the manuscript.
5. Were the IC50 of Rottlerin on water permeability of hAQP1 and hAQP3 and on glycerol permeability of hAQP3 determined using dose-response curves?
Reply: Yes, the IC50 values were obtained using dose-response curves in hRBCs, as shown in Fig 2D (water permeability) and Fig 2E (glycerol permeability). In yeast cells we validated RoT selectivity towards AQP3 using a higher concentration of RoT (100 μM), to allow detecting any effect in AQP1 that would only be visible with larger concentrations of the compound.
6. Are the doses of Rottlerin used to inhibit AQPs permeabilities in red blood cells and yeast compatible with cell usage (are they inducing any cell mortality) and do they inhibit protein kinase C activity as well?
Reply: The doses used were not toxic to the cells, since no differences in the number of cells in suspension previous to the assays were observed. Cell death would be detected by RBC hemolysis and a quenching of the stopped-flow signal due to the lower number of responsive cells, both in RBCs and yeast assays, which was not the case in our experiments. We assured that the number of viable cells was not affected during the experiment.
The permeability assays are performed within a very short time scale: 5 seconds data acquisition for water permeability assays, 30-40 seconds data acquisition for glycerol permeability assays. The possible effect of RoT on Prot Kinase C would not affect our permeability data within this short time frame. Moreover, inhibition of Prot Kinase C by RoT would result in a lower AQP phosphorylation and translocation to the membrane, but our cells already have established levels of AQP expression at the membrane previous to the assays, as demonstrated in the new Figure S1 Supplemental Material (GFP-tagged hAQP1 and hAQP3 in yeast cells).
7. Does Rottlerin affect the water and glycerol permeabilities of other AQPs?
Reply: Thank you for the very relevant comment. This study was focused on the effect of RoT on AQP1 and AQP3 that are considered important drug targets for cancer. In a future study we will certainly explore the possible effect of RoT on other AQPs, investigating mechanisms of selectivity and structure activity relationships.
8. Protein kinase C has been involved in cell volume regulation (e.g. PMIDs: 9755245, 8038218, 1560240) as well as phosphorylation and translocation of AQP1 (PMID: 22334691). Beyond the computational analysis, what are the experimental evidence for a direct effect of Rottlerin on hAQP3, rather than an indirect effect via its inhibitory action on protein kinase C that may be stimulated upon osmotic challenge?
Reply: Protein kinase C may affect translocation and phosphorylation of AQP1. However, as demonstrated in the new Figure S1 (GFP-tagged hAQP1 and hAQP3 in yeast cells), AQP1 is already localised at the plasma membrane previous to permeability assays, assuring that the AQP1 channel is well expressed, localised, and is functional (validated with control cells without treatment). Moreover, in this study AQP1 permeability was not inhibited, thus discarding any effect of other possible inhibition mechanisms.
As for AQP3, the protein is also localised at the membrane and functional (Fig. S1 and Fig. 3C and D). Upon osmotic challenge AQP3 inhibition is measured within a very short time scale (5 sec for water and 30-40 sec for glycerol). In case Protein Kinase C would be stimulated it would affect AQP3 translocation pathway, upstream the protein insertion in membrane; this would not be reflected in our permeability assays that measure directly the channel activity, a few seconds after the osmotic challenge.
9. What are the effects of other protein kinase C inhibitors on the hAQP1 and hAQP3 water and glycerol permeabilities? Therefore, in cancers, the authors should discuss if the effects of Rottlerin may be attributable to protein kinase C and/or AQP3 inhibition, taking into account protein kinase C has also been involved in cancers.
Reply: Protein Kinase C may affect phosphorylation and translocation of proteins to the membrane, and this is surely an important factor in cancer. Our study evaluated the activity of the channel that is already localised at the membrane before the assays, and we used direct measurements of membrane permeability with appropriate controls for expression and for function. We addressed this point and inserted a sentence in the discussion to clarify: “The confirmed AQP3 expression and localization at the cell plasma membrane previous to functional assays indicates that RoT has a direct effect on AQP3 channel activity with no interference in upstream protein translocation mediated by Protein Kinase C. ” The effect of other Protein kinase C inhibitors on the expression of AQPs is beyond the scope of this study, although it may deserve future investigation.
Reviewer 3 Report
In the present study, Alves et.al have studied the inhibitory effect of rottlerin on aquaporin-3 using experimental and computational techniques. Please see my comments below:
- “initially generate a sequence amino acid alignment of AQP3 with AQP10, and afterward 404 generate 1000 different models of AQP3 protein monomer”
Authors should mention in the method section how they transform monomer to tetrameter after performing comparative modeling.
- “To integrate the chosen model on a pure POPC bilayer membrane”
The authors should mention, the source from where the POPC bilayer was taken. Did the authors equilibrate the bilayer before performing simulations with protein?
- “delete lipids within a 418 user-defined distance cutoff around the protein”
The authors should mention what cut-off they used in this study.
- “pKa, determined using the pyPKA server”
pyPKA server or python tool?
5. “The topology of 449 RoT was obtained from the Automated Topology Builder and Repository [62,63], and 450 manually curated to include the complete 1–4 exclusions in rings”
Authors should cite the work from which they have taken 1-4 exclusions parameters for the rings.
Author Response
Thank you for the careful evaluation of our manuscript. Please see below the replies to the raised questions.
- In the present study, Alves et.al have studied the inhibitory effect of rottlerin on aquaporin-3 using experimental and computational techniques. Please see my comments below:
1. “initially generate a sequence amino acid alignment of AQP3 with AQP10, and afterward 404 generate 1000 different models of AQP3 protein monomer”.
Authors should mention in the method section how they transform monomer to tetrameter after performing comparative modeling.
Reply: We thank the referee for the suggestion. As described in the manuscript, the process of building up the tetrameric form of AQP3 was done by initially generating the monomeric version of the protein. Afterward, based on the tetrameric version of the AQP10 (crystallographic structure used as the reference structure to generate AQP3 model), we have fitted 4 individual monomers of AQP3 on the corresponding monomers of AQP10 protein. With this approach, we were able to generate a 3D model of the full tetrameric version of AQP3. The used procedure is explained at the end of section 2.7: “The full homotetramer AQP3 structure was then reconstructed based on AQP10 homotetrameric structure, by aligning the generated AQP3 monomer, to each one of the four monomers from the template structure. In the end, the full tetrameric structure of AQP3 was obtained (new Figure S2).”
2. “To integrate the chosen model on a pure POPC bilayer membrane”
The authors should mention, the source from where the POPC bilayer was taken. Did the authors equilibrate the bilayer before performing simulations with protein?
Reply: We thank the reviewer for pointing out the missing information. Indeed we have used a previously pre-equilibrated POPC membrane bilayer (Magalhães et J Chem Inf Model 2022, 62, 3034-3042, doi:10.1021/acs.jcim.2c00372) to integrate the AQP3 tetrameric structure. We have added this detail to the methods section of the manuscript.
3. “delete lipids within a 418 user-defined distance cutoff around the protein”
The authors should mention what cut-off they used in this study.
Reply: In the inflateGRO2 protocol used in this work, we have set up a 5 angstroms cut-off between the lipids and the protein atoms. We have added this missing information to the manuscript.
4. “pKa, determined using the pyPKA server”
pyPKA server or python tool?
Reply: Although we could have used the pyPKA python tool, we decided to use its implementation in the pyPKA server, which can be found at https://pypka.org/run-pypka/ . This is correctly mentioned in section 2.8: “All titrating residues of the protein were assigned with an initial protonation state determined at physiological pH, based on their pKa, determined using the pyPKA server.”
5. “The topology of 449 RoT was obtained from the Automated Topology Builder and Repository [62,63], and 450 manually curated to include the complete 1–4 exclusions in rings”
Authors should cite the work from which they have taken 1-4 exclusions parameters for the rings.
Reply: As pointed out by Malde et al. (J Chem Theory Comput 2011, 7, 4026-4037, doi:10.1021/ct200196m), molecules from the ATB repository may have an incomplete description of the 1-4 exclusions found in the rings of the parameterized molecules. This was the case for RoT, and therefore we have manually checked the topology and added the required missing 1-4 exclusions parameters. To do so, we used the same approach used in previous works from our group (Gaspar et al., Forensic Sci Int 2018, 290, 146-156, doi:10.1016/j.forsciint.2018.07.001) and (Magalhães et J Chem Inf Model 2022, 62, 3034-3042, doi:10.1021/acs.jcim.2c00372). We have completed this information in the manuscript, by including works where this approach was previously used.
Round 2
Reviewer 1 Report
The references about the localization and functions of AQP1 and AQP3 on human red cells should be cited in the manuscript.
Author Response
The references were cited in the text according to the reviewer suggestion.
Reviewer 2 Report
-
Author Response
Thank you.
Reviewer 3 Report
The authors have made changes suggested by me, therefore, I would recommend to accept this manuscript.
Author Response
Thank you.